# Diagnosis and Outcomes of Fungal Co-Infections in COVID-19 Infections: A Retrospective Study

**DOI:** 10.3390/microorganisms11092326

**Published:** 2023-09-15

**Authors:** Richard Swaney, Rutendo Jokomo-Nyakabau, Anny A. N. Nguyen, Dorothy Kenny, Paul G. Millner, Mohammad Selim, Christopher J. Destache, Manasa Velagapudi

**Affiliations:** 1Internal Medicine/Infectious Diseases, Creighton University School of Medicine, Omaha, NE 68178, USA; richardswaney@creighton.edu; 2Internal Medicine, Creighton University School of Medicine, Omaha, NE 68178, USA; rutendojokomonyakabau@creighton.edu (R.J.-N.); paulmillner@creighton.edu (P.G.M.); mohammadselim@creighton.edu (M.S.); 3School of Medicine, Creighton University, Omaha, NE 68178, USA; annynguyen@creighton.edu (A.A.N.N.);; 4School of Pharmacy & Health Professions, Creighton University, Omaha, NE 68178, USA; chrisdestache@creighton.edu; 5Division of Infectious Diseases, Catholic Health Initiative (CHI) Health Creighton University Medical Center, Omaha, NE 68178, USA

**Keywords:** COVID-19, diagnosis, fungal infection, outcome

## Abstract

The SARS-CoV-2 pandemic has resulted in a public health emergency with unique complications such as the development of fungal co-infections. The diagnosis of fungal infections can be challenging due to confounding imaging studies and difficulty obtaining histopathology. In this retrospective study, 173 patients with COVID-19 receiving antifungal therapy due to concern for fungal co-infection were evaluated. Patient characteristics, clinical outcomes, and the utility of fungal biomarkers were then evaluated for continuation of antifungal therapy. Data were collected from the electronic health record (EPIC) and analyzed using SPSS (version. 28, IBM, Inc., Armonk, NY, USA) Data are presented as mean ± SD or percentages. A total of 56 COVID-19 patients were diagnosed with fungal co-infection and 117 COVID-19 + patients had no fungal infection. Significantly fewer female patients were in the fungal+ group compared to COVID-19 control patients (29% in fungal+ compared to 51% in controls *p* = 0.005). Fungal diagnostics were all significantly higher in fungal+ patients. These include 1,4-beta-D-glucan (BDG), fungal culture, and bronchoalveolar lavage galactomannan (BAL GM). Intensive care unit hospitalization, mechanical ventilation, and mortality in fungal+ patients with COVID-19 were significantly higher than in control patients. Finally, significantly more fungal+ patients received voriconazole, isavuconazonium, or amphotericin B therapies, whereas control patients received significantly more short-course fluconazole. COVID-19+ patients with fungal co-infection were significantly more likely to be in the ICU and mechanically ventilated, and they result in higher mortality compared to control COVID-19 patients. The use of fungal diagnostics markers were helpful for diagnosis.

## 1. Introduction

The SARS-CoV-2 pandemic has resulted in a public health emergency of global concern with a multitude of complications and morbidities. While the effects and consequences of bacterial and viral co-infections have been reported extensively in the literature, there is less understanding of fungal co-infections in patients with COVID-19 [1,2]. The most common fungal infection appears to be COVID-19-associated pulmonary aspergillosis (CAPA), but there have been reports of other fungal co-infections, including, but not limited to, invasive candidiasis, mucormycosis, pneumocystis pneumonia, histoplasmosis, and cryptococcosis [3]. The reported rate of these infections varies greatly between different studies. For CAPA, the prevalence may range anywhere from 3 to 33%; a recent multicenter study reported a prevalence ranging from 1.7 to 26.8% in ICU patients across all centers studied. In comparison, invasive pulmonary aspergillosis was found in up to 15% of clinical and autopsy studies of immunocompetent patients who developed acute respiratory distress syndrome (ARDS) [4]. The incidence of candidemia in COVID-19 patients has been reported as being between 5 and 9% [5,6]. A study in Egypt reported an incidence rate of 7.63% for COVID-19-associated mucormycosis, although incidence is thought to be significantly higher in India [7,8]. Secondary fungal infections are speculated to occur due to the direct impact of SARS-CoV-2 in decreasing the immune response, as well as due to sepsis disrupting the mucosal barriers and sequelae of common COVID-19 treatments such as corticosteroids and prolonged mechanical ventilation [9]. Mortality has been reported to be higher in patients with fungal co-infections such as CAPA, indicating the importance of prompt diagnosis and treatment [10]. One study reported that mortality risk in CAPA patients was almost twice that of non-CAPA patients in the ICU setting [3].

The diagnosis of COVID-19 fungal co-infections can be challenging, as common risk factors seen with invasive fungal disease such as neutropenia, immunosuppression, or malignancy may be absent [11]. Another challenge is the lack of a standardized diagnostic algorithm for co-fungal infections. For example, while CAPA is one of the more well-studied infections, there is still no definitive approach to diagnosing CAPA, and most of the criteria are still speculative [12]. For CAPA, diagnosis is recommended using the presence of clinical features in combination with mycological lab evidence, including aspergillus growth in fungal culture, positive PCR, beta-D-glucan (BDG), and serum or bronchoalveolar lavage (BAL), for detection of *Aspergillus* galactomannan (GM). Bronchoscopy is needed for definitive diagnosis of CAPA; however, concerns of virus aerosolization limited the use of this procedure during the early stages of the pandemic [13]. Although tissue biopsy is considered the gold standard for definitive diagnosis of fungal infection, it is less commonly performed due to the risk of patient complications [10] Sputum, BAL, or tracheal aspirate cultures can be utilized, but carry risks of contamination by upper respiratory flora and possible false positivity [14,15].

Fungal biomarkers such as serum and BAL GM and serum BDG are commonly used but present challenges as well. For instance, serum and BAL GM have shown reduced sensitivities when applied to non-neutropenic COVID-19 patients [13], with one study indicating a sensitivity of 46.9% and 54.5% for serum and BAL GM, respectively [16]. Another problem with BAL GM is its inability to distinguish between *Aspergillus* colonization and actual infection. Serum BDG might exhibit higher sensitivity than GM, but it lacks specificity as it rises in various fungal infections and fails to identify the responsible organisms [17]. These observations underscore the necessity for more comprehensive research and assessment of the diagnostic capabilities of fungal biomarkers in detecting fungal co-infections in COVID-19 patients.

In this study, the effectiveness of fungal biomarkers such as fungal culture, serum BDG, and serum and BAL GM in diagnosing fungal co-infections in COVID-19 patients was assessed along the clinical outcomes of COVID-19 patients with fungal co-infections.

## 2. Methods

The study was a retrospective, observational study examining patients admitted to hospital with COVID-19 diagnosis who received antifungal therapy. The study population were patients admitted to CHI Health Nebraska hospital centers (inpatient hospital facilities located in Omaha, NE, USA; Council Bluffs, IA, USA; Lincoln, NE, USA; and Grand Island, NE, USA) from June 2020 to December 2021. Inclusion criteria included patients olds than 19 years of age with positive COVID-19 diagnosis through PCR (polymerase chain reaction) testing who received antifungal agents of either fluconazole, voriconazole, isavuconazonium, amphotericin B, micafungin, or itraconazole during their hospitalization. Data were then divided into COVID-19 patients with diagnostic evidence for development of a fungal infection (fungal+) and COVID-19 patients without diagnostic/clinical evidence for fungal infection (controls). Patients were categorized as fungal+ if they had positive fungal biomarker testing, (BDG, *cryptococcus* antigen, urine, and serum *histoplasma* antigen, *histoplasma* antibody, BAL *Aspergillus* GM or positive blood, sputum, or BAL cultures for fungal elements) and received continued treatment for invasive fungal disease after diagnosis. Patients with sputum or BAL cultures positive for *Candida* species were excluded from the positive fungal culture group. Control patients were categorized as those with COVID-19 whose empiric antifungal therapy was discontinued based on negative testing, sputum, BAL cultures positive for candida species only, or lack of clinical parameters for fungal disease.

Patient data were collected from the electronic medical record (EPIC),including demographics (age, gender, and race (Hispanic, African American, White, or Asian)), comorbidities (BMI, history of malignancy, history of solid organ transplant, high dose corticosteroid use 4 weeks prior to admission, use of chronic immunosuppressant therapy, and HIV status), length of hospitalization, need for ICU admission, need for mechanical intubation, need for extracorporeal membrane oxygenation (ECMO), COVID-19 treatment (corticosteroids, remdesivir, tocilizumab, or barcitinib), fungal treatment (voriconazole, posaconazole, isavuconazonium, amphotericin B, fluconazole, micafungin, itraconazole or other antifungal therapy). Patient outcome (treated or expired) was also collected.

Data were collected in Excel and analyzed using SPSS (ver. 28, IBM, Inc., Armonk, NY, USA).

Descriptive data were analyzed using Chi-square or Fisher’s exact test. Continuous data were analyzed with unpaired Student *t*-test. *Apriori* level of significance was *p* ≤ 0.05.

## 3. Results

A total of 177 patients were identified as having a positive COVID-19 diagnosis and use of antifungal therapy during their hospitalization. Four patients were identified as having fungal infections prior to hospitalization for COVID and were thus excluded, resulting in 173 patients in the study. Based on positive fungal testing in which the primary team determined antifungal treatment was necessary, 56 patients were categorized in the fungal+ group. Those categorized as the control group, whose antifungal therapy was discontinued either from negative fungal testing or lack of physician clinical suspicion, included 117 patients.

Table 1 lists relevant demographic data between the two groups. Significantly fewer female patients were in the fungal+ group compared to COVID-19+ control patients (29% in fungal+ compared to 51% in control, *p* = 0.005); however, age, BMI, race, and history of malignancy, neutropenia, or recent immunosuppression use were not significant between the two groups.

Table 2 outlines the comparison between the fungal+ group and the control group in hospital management and outcomes. Significantly more fungal+ COVID-19 patients were hospitalized in the intensive care unit compared to control patients (95% in fungal+ compared to 68% in control, *p* < 0.001). Significantly more fungal+ patients were mechanically ventilated compared to control patients (91% compared to 54%, *p* < 0.001). The duration of mechanical ventilation averaged 4.5 days longer in fungal+ patients but did not reach statistical significance. Significantly more fungal+ patients underwent bronchoscopy compared to control patients (64% compared to 34%, *p* < 0.001). Finally, the mortality rate in fungal+ patients with COVID-19 was significantly higher (63%) compared to the control COVID-19 patients (42%, *p* = 0.010). There were five control patients and three fungal+ patients who received ECMO therapy. Treatment of COVID-19 with remdesivir and immune modulators were not significantly different between the two groups of patients; however, all fungal+ patients with COVID-19 received corticosteroids compared to 89% in the control group, *p* = 0.01. Significantly more fungal+ patients received concurrent antimicrobial therapy compared to control patients (98% in fungal+ compared to 83% in control, *p* = 0.002).

Fungal diagnostics were all significantly higher in fungal+ patients. These include BDG, fungal culture and BAL GM. Fungus species isolated from positive cultures are available in Table 3.

Fungal culture positivity was significantly different between the fungal+ and control groups. Of the patients in the fungal+ group with positive fungal cultures (73%), aspergillus was the most frequently isolated organism (66%), followed by *Candida* blood stream infections (22%). *Cryptococcus neoformans* was isolated in two cases, while *Histoplasma* and *Fusarium* was isolated in one case each. In one instance, a mold was isolated which was not able to be further identified. Other diagnostic tests, including *cryptococcal* antigen, serum and urinary *histoplasma* antigen, and *histoplasma* antibody, were not performed in both groups of COVID-19 patients.

Fungal+ patients received significantly more voriconazole, isavuconazonium, or amphotericin B therapies, whereas control patients received significantly more fluconazole. Significantly more fungal+ patients were diagnosed with CAPA (50%), whereas none of the control patients received this diagnosis, *p* < 0.001. Finally, length of hospitalization averaged 27 days in fungal+ patients and was not different from COVID-19 patients without fungal infections averaging 25 days.

## 4. Discussion

According to the existing literature, the simultaneous presence of fungal co-infection in individuals with COVID-19 poses a significant danger, particularly for those with pre-existing conditions. This can result in the worsening of complications and ultimately lead to a higher mortality rate [18]. The virus is known to cause immune dysregulation, an overproduction of pro-inflammatory cytokines, a weakened cell-mediated immunity, and a decrease in CD4 and CD8+ T-cells, all of which can increase the likelihood of invasive fungal infections [19,20,21]. Fungal co-infections in patients with COVID-19 have non-specific imaging findings. These patients are at high risk of progression to ARDS and bacterial infections that can mimic fungal infections [22], making the diagnosis challenging.

Here, 56 patients who were treated with COVID-19 and fungal co-infections were compared with patients who did not have a fungal co-infection. This study used a variety of fungal biomarkers, including serum BDG and BAL GM, and respiratory cultures to help confirm fungal infections in suspected populations with COVID-19 infections. While serum BDG is a non-specific test with high negative predictive value, the risk of false positivity can deter physicians from using it to accurately diagnose and treat patients for fungal co-infections [23]. Our study demonstrated a statistically significant correlation in elevated BDG assays and the decision to treat patients for invasive fungal infections. In the fungal+ group, however, BDG was not the only positive fungal biomarker when deciding to treat fungal infections. All patients in the fungal+ group with positive BDG also had either a BAL GM or were fungal culture-positive before deciding to continue treatment. On the other hand, the control group had seven patients with positive BDG assay without any other positive biomarkers. In these patients, the decision was made to stop antifungal therapy. This study supports the role of BDG as an adjunctive test for determining fungal co-infection in COVID-19 patients.

While diagnosis of confirmed fungal infections requires histopathological diagnosis, this was not performed in this study and is not frequently used in real-world settings. Fungal cultures are a strong alternative to histopathological testing and allow for less invasive bronchoscopic procedures with less risk of complications. Our study demonstrated that positive fungal cultures allowed physicians to confidently diagnosis and treat patients of COVID-19 with fungal co-infections. Furthermore, the fungal+ group had positive fungal cultures in addition to the fungal biomarkers. Using these tests together allowed for a higher degree of suspicion of true infection, i.e., rather than fungal culture alone, in determining true infection versus colonization/false positivity when treating the patient group. This study highlights the importance of obtaining fungal culture to guiding the decision to treat patients with fungal co-infections.

Of the patients in the fungal+ group with positive fungal cultures, *Aspergillus* was the most frequently isolated organism (n = 27), followed by *Candida* fungemia (n = 9). This study is consistent with other studies which outline aspergillosis as the most common fungal co-infection in COVID-19 infections, followed by candidemia. Of note, two cases of identified *Cryptococcal* fungemia were found. The association between COVID-19 infections and *Cryptococcal* fungemia has not been widely reported and warrants future studies. Although COVID-19-associated mucormycosis have been identified in the literature, none were identified on our retrospective study.

First line treatment for invasive aspergillosis infections includes voriconazole or isavuconazonium. Primary treatment for invasive candidemia includes fluconazole and echinocandins. Amphotericin B is usually reserved for severe infections refractory to primary treatments due to the severe side-effect profile. Comparison of antifungal agents used in the fungal+ group versus the control group demonstrated a significantly increased use of voriconazole, isavuconazonium, and amphotericin B in the fungal+ group and a significantly increased use of fluconazole for the control group. This discrepancy in antifungal agents used between groups could be due to concern for the adverse effects of voriconazole and amphotericin B, in which case their use would be limited in patients with confirmed diagnosis only through positive cultures, or in patients with higher degrees of suspicion. Conversely, the increased use of fluconazole may be due to lower concern for infection, less severe adverse effects, and their potential use for non-invasive fungal infections such as vulvovaginal candidiasis or oral thrush.

Immune modulating therapy is an important risk factor in the development of fungal infections. Corticosteroids, which are recommended in COVID-19 treatment, pose a risk of immunosuppression and the development of fungal infections. Previous studies have identified corticosteroid use in patients with severe influenza infections as a risk for the development of invasive pulmonary aspergillosis [24]. Furthermore, immunomodulators are recommended in COVID-19 treatment to help reduce the hyperactive inflammatory response [25]. The risk of these immunomodulators for invasive fungal infections is of interest. Our study looked at the use of corticosteroids, baricitinib, and tocilizumab in patients treated with COVID-19 and found no statistically significant increase in risk with respect to positive fungal testing and treatment for COVID-19 fungal co-infection.

A limitation of our study was its lack of insight into the doses and duration of these immunosuppressive medications. Further research into the dosing of corticosteroids and the risk of developing COVID-19 fungal co-infections could be a useful area of research in the future.

Another limitation was the retrospective nature of the study. Further prospective studies should be performed to help remove sampling bias.

## 5. Conclusions

Prompt recognition with the use of fungal biomarkers and treatments for fungal co-infection with COVID19 is key to reducing delays in diagnosis and treatment to prevent complications and death from these infections. In cases of COVID-19 and fungal co-infection, the likelihood of positive fungal biomarkers such as BDG and GM was higher. Although the use of these biomarkers for diagnosis was not common among COVID-positive patients, those who did undergo testing were found to have a greater chance of testing positive for fungal infection. In such cases, fungal culture was often used to prompt antifungal therapy, with voriconazole treatment being the most common course of action.

## Figures and Tables

**Table 1 microorganisms-11-02326-t001:** Study demographics.

Variable Mean ± SD or No (%)	Controls (N = 117)	Fungal+ (N = 56)	*p*-Value
Age (yrs)	58.7 ± 14.9	62.6 ± 11.5	NS
BMI (m^2^/kg)	33.9 ± 11.2	33.2 ± 7.4	NS
Females	60 (51)	16 (29)	0.005
Race			NS
Hispanic	15 (13)	4 (7)	NS
Black	11 (9)	3 (5)	NS
White	87 (74)	43 (77)	NS
Asian	1 (1)	0 (0)	NS
Other	2 (2)	3 (5)	NS
Cancer history	12 (10)	4 (7)	NS
Neutropenia	2 (1)	1 (2)	NS
Immunosuppressive therapy	10 (9)	6 (11)	NS

BMI = body mass index; NS = not significant; SD = standard deviation.

**Table 2 microorganisms-11-02326-t002:** Comparison of study results.

Variable Mean ± SD or No (%)	Controls (N = 117)	Fungal+ (N = 56)	*p*-Value
ICU bed	80 (68)	53 (95)	<0.001
Mechanical ventilation	63 (54)	51 (91)	<0.001
Bronchoscopy procedure	42 (34)	36 (64)	<0.001
Mortality rate	51 (42)	35 (63)	0.010
Antimicrobial therapy	97 (83)	55 (98)	0.002
Corticosteroid use	105 (89)	56 (100)	0.01
Remdesivir use	87 (74)	48 (86)	NS
Tocilizumab use	24 (21)	18 (32)	NS
Baricitinib use	28 (24)	10 (18)	NS
Fungal diagnostics			
Beta-D-glucan (BDG)	7 (6)	19 (34)	<0.001
Culture	3 (6)	41 (73)	<0.001
BAL Galactomannan (GM)	1 (2)	28 (50)	<0.001
Fungal treatment			
Voriconazole	35 (30)	45 (80)	<0.001
Isuvoconazole	1 (1)	5 (9)	<0.007
Amphotericin B	1 (1)	5 (9)	<0.007
Fluconazole	59 (50)	8 (14)	<0.001
Micafungin	27 (23)	10 (18)	NS
Others	15 (13)	5 (9)	NS
CAPA diagnosed	0 (0)	28 (50)	<0.001
Length of hospitalization (d)	25.6 ± 20.1	27 ± 17.9	NS

NS = not significant; CAPA = COVID-19-associated pulmonary aspergillosis; BAL = bronchoalveolar lavage; ICU = intensive care unit.

**Table 3 microorganisms-11-02326-t003:** Fungal culture identification.

Fungal Pathogen Cultured	No. (%)
*Aspergillus* species	27 (65%)
*Candida* species on blood culture	9 (21%)
*Cryptococcus*	2 (1%)
*Fusarium*	1 (<1%)
*Histoplasmosis*	1 (<1%)
Other (mold not identified)	1 (<1%)

## Data Availability

To interested parties upon request.

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
