# Peer review of "Diagnosis and Outcomes of Fungal Co-Infections in COVID-19 Infections: A Retrospective Study"

_microorganisms, 2023, doi:10.3390/microorganisms11092326_

Round 1

Reviewer 1 Report

Dear authors, I would like to congratulate you for the efforts conducted when performing the present study.

Here goes a few concerns:

The Abstract does not require sub-heading according to the authors guidelines.

I recommend the authors to place the keywords by alphabetic order.

The citation in the text should follow the journal format.

The abbreviation in the manuscript should have an extended name associated to it the first time they appear in the text. Please review the abbreviations in the manuscript.

The study rational, prior to the aim sentence should be somehow improved.

The aim question, in the end of the Introduction, should be more clear regarding which aspects are to be assessed and which outcomes to be evaluated.

How were the patients selected? All included? Random?

Provide the meaning of SD in the footnote of Table 1 and 2.

Please debate the strength of the study.

Please debate the internal and external validity of the study.

Please be aware that the final list of references is not following the journal guidelines.

Author Response

1.The Abstract does not require sub-heading according to the authors guidelines.

  • Thank You for pointing this out. We agree with this comment.
  • Therefore, we have deleted the sub-headings in the abstract .

2.I recommend the authors to place the keywords by alphabetic order.

  • Thank You for pointing this out.
  • We agree with this comment. The keywords are now listed in alphabetical order.

3.The citation in the text should follow the journal format.

  • Thank You for pointing this out.We agree with this comment.
  • We have edited all citations as per the journal format.

4.The abbreviation in the manuscript should have an extended name associated to it the first time they appear in the text. Please review the abbreviations in the manuscript.

  • Thank You for pointing this out. We agree with this comment.
  • We verified and made sure all the abbreviations have an extended name the first time they were used.

5.The study rational, prior to the aim sentence should be somehow improved.

Thank You for pointing this out. We agree with this comment.

  • It has been rewritten to explain the rationale of diagnostic dilemmas as below:
  • Fungal biomarkers, such as serum and BAL GM and serum BDG, are commonly used but present challenges as well. For instance, serum and BAL GM have shown reduced sensitivities when applied to non-neutropenic COVID-19 patients [13], with one study indicating a sensitivity of 46.9% and 54.5% for serum and BAL GM respectively [16]. Another problem with BAL GM is its inability to distinguish between Aspergillus colonization and actual infection. Serum BDG might exhibit higher sensitivity than GM, but it lacks specificity as it rises in various fungal infections and fails to identify the responsible organism [17]. These observations underscore the necessity for more comprehensive research and assessment of the diagnostic capabilities of fungal biomarkers in detecting fungal co-infections in COVID-19 patients.

6.The aim question, in the end of the Introduction, should be clearer regarding which aspects are to be assessed and which outcomes to be evaluated.

  • Thank You for pointing this out. We agree with this comment.
  • The last paragraph in the introduction has been modified to improve the reason for the study rationale as below.
  • In this study, the effectiveness of fungal biomarkers such as fungal culture, serum BDG, and serum and BAL GM in diagnosing fungal co-infections in COVID-19 patients, was assessed and the clinical outcomes of COVID-19 patients with fungal co-infections.

7.How were the patients selected? All included? Random?

  • Thank You for pointing this out. We agree with this comment.
  • Patients with covid -19 and on antifungal therapy were reviewed. Out of those, patients with clinical suspicion of fungal infections were selected as cases and those with suspicion but workup negative for fungal infection and fungal infection rule out were selected as controls.

8.Provide the meaning of SD in the footnote of Table 1 and 2.

  • Thank You for pointing this out. We agree with this comment.
  • Defined the abbreviation SD as “Standard deviation “ in the foot note

9.Please debate the strength of the study.

  •  Thank You for pointing this out. We agree with this comment. Please find the response below
  •  The study revealed the complex challenges of diagnosing fungal infections, especially in the context of COVID-19, where difficulties in interpreting imaging and limitations in histopathology were identified. Diagnostic markers for fungal infections, such as 1,4 beta-D-glucan (BDG), fungal culture, and bronchoalveolar lavage galactomannan (BAL GM), were significantly higher in patients with fungal co-infections. The study also highlighted differences in the use of antifungal treatments between these two groups. Importantly, this study emphasizes the crucial role of accurate fungal diagnostics in identifying and effectively managing fungal co-infections in COVID-19 patients, as well as their significant impact on clinical outcomes.

10. Please debate the internal and external validity of the study.

  • Thank You for pointing this out. We agree with this comment.
  •  Internal validity for these study results are moderate as there was no randomization or comparison of antifungal efficacy.  Participants demographics (except female sex) and severity of illness (hospitalization length) did not differ substantially.  However, the passage of time could influence the results as additional COVID-19 treatment modalities became available later in the study as compared to early in the study including antibiotic therapy (significantly higher in controls) and corticosteroid use (significantly higher in Fungal+ group).  No differences in study groups for COVID-19 treatment modalities.  External validity of these study results is high as these results are similar to real-world treatment of fungal infections after pulmonary injury from viral illness.  Thus, the authors believe that other study results would be like these study results. 

11.Please be aware that the final list of references is not following the journal guidelines.

  • Thank You for pointing this out. We agree with this comment.
  • References have been edited as per journal guidelines now.

Reviewer 2 Report

The manuscript entitled (Diagnosis and Outcomes of Fungal Co-infections in COVID-19 infections: A Retrospective study) by Swaney et al. reported the utility of fungal biomarkers including fungal culture, serum BDG, and serum and BAL GM was examined, in the diagnosis of fungal co-infection with COVID-19, as well as the clinical outcomes of these patients. Effectiveness of antifungal drugs as voriconazole, isavuconazonium, or amphotericin B in treatment of COVID-19. The study revealed that the use of anti-fungal therapy voriconazole being the most common course of action. The manuscript is good and can be accepted after the following revision.

1-      In key words the word `` fungal infection`` should written ``Fungal infection``.

2-      The ethical approval for the patients and from Hospital should be provided in the MS.

3-      Tables 1 and 2 should be drawn as graph.

4-      Different abbreviations especially in the discussion part should be written, please check all abbreviations throughout the whole MS.

5-      The data for the following: Author Contributions/Funding/Institutional Review Board Statement/Informed Consent Statement/ Data Availability Statement/Conflicts of Interest should be completed.

6-      Some fungi names should be written in italic form as cryptococcal fungemia, please check all MS.

7-      Are there any additional side effects observed during the treatment with antifungal drugs between the patients more the normal and reported side effects for antifungal drugs???? If yes please clarify.

Minor editing of English language required

Author Response

  • In key words the word `` fungal infection`` should written ``Fungal infection``.
  • Thank You for bringing that to my attention.
  • It has been changed to “Fungal Infection”

2-      The ethical approval for the patients and from Hospital should be provided in the MS.

  • Thank You for pointing this out. We agree with this comment.
  • Given the retrospective design of the study, the institutional IRB waived the need for consent

3-Tables 1 and 2 should be drawn as graph.

  • Thank You for the comment.
  • The authors feel that Tables 1 and 2 should be represented as tables instead of graphs as the number of bars for a graph with percentages on the Y-axis would be too confusing for readers.

4-      Different abbreviations especially in the discussion part should be written, please check all abbreviations throughout the whole MS.

  • Thank You for pointing this out. We agree with this comment.
  • Checked all abbreviations and made necessary edits

5-      The data for the following: Author Contributions/Funding/Institutional Review Board Statement/Informed Consent Statement/ Data Availability Statement/Conflicts of Interest should be completed.

  • Thank You for pointing this out. We agree with this comment.
  • All the requested data included now

6-      Some fungi names should be written in italic form as cryptococcal fungemia, please check all MS.

  • Thank You for pointing this out. We agree with this comment.
  • We have edited to write all fungi names in italics

7-      Are there any additional side effects observed during the treatment with antifungal drugs between the patients more the normal and reported side effects for antifungal drugs???? If yes please clarify.

  • Thank You for pointing this out.
  • Data regarding the adverse effects of the antifungals was not collected

Reviewer 3 Report

This retrospective analysis, involving 173 individuals with COVID-19, delved into the realm of fungal co-infections and their ramifications. The study illuminated the formidable intricacies of diagnosing fungal infections, particularly within the context of COVID-19, where challenges stemming from perplexing imaging interpretations and histopathology limitations were discerned. Fungal diagnostic markers, namely 1,4 beta-D-glucan (BDG), fungal culture, and bronchoalveolar lavage galactomannan (BAL GM), were notably elevated in patients afflicted by fungal co-infections. Unveiling a concerning trend, COVID-19 patients contending with fungal co-infections exhibited elevated frequencies of intensive care unit admissions, reliance on mechanical ventilation, and increased mortality rates in comparison to those without fungal involvement. Distinct utilization of antifungal treatments also emerged between these two cohorts. Notably, the study underscores the paramount importance of precise fungal diagnostics in not only identifying and effectively managing fungal co-infections among COVID-19 patients, but also in profoundly influencing their clinical trajectories.

Nonetheless, I would like to bring forth a few inquiries for the researchers to address. Firstly, could you expound on the rationale behind confining the study's timeline solely to the period spanning from June 2020 to December 2021? It would be prudent to consider incorporating a schematic representation within the methodology section, elucidating the procedural workflow. Additionally, given the association between COVID-19 and mucormycosis, I am curious if any instances of mucor infection were documented within the study's participant pool. Is there any available data delineating the potential impacts of vaccination on fungal co-infection within individuals afflicted by COVID-19? Moreover, considering the ever-evolving landscape of circulating variants of concern, have any investigations been conducted to ascertain their potential influence on fungal co-infections among COVID-19 patients within the study period?

Author Response

1.Firstly, could you expound on the rationale behind confining the study's timeline solely to the period spanning from June 2020 to December 2021?

  • Thank You for pointing this out.
  • We selected that period as it was during that time we had maximum incidence of fungal co-infections in COVID-19 patients at our institution

2.It would be prudent to consider incorporating a schematic representation within the methodology section, elucidating the procedural workflow.

  • Thank You for the comment
  • The authors believe that the methods describe the procedural workflow appropriately and a schematic defeat this.
  1. Additionally, given the association between COVID-19 and mucormycosis, I am curious if any instances of mucor infection were documented within the study's participant pool.
  • No cases of co-infection with mucormycosis were noted in our study population

4.Is there any available data delineating the potential impacts of vaccination on fungal co-infection within individuals afflicted by COVID-19?

  • Thank You for pointing this out. We agree with this comment.
  • This was not studied

Round 2

Reviewer 1 Report

Dear author, I have no further comments.